# Elucidating the Role of Innate and Adaptive Immune Responses in the Pathogenesis of Canine Chronic Inflammatory Enteropathy—A Search for Potential Biomarkers

**DOI:** 10.3390/ani12131645

**Published:** 2022-06-27

**Authors:** Daniela Siel, Caroll J. Beltrán, Eduard Martínez, Macarena Pino, Nazla Vargas, Alexandra Salinas, Oliver Pérez, Ismael Pereira, Galia Ramírez-Toloza

**Affiliations:** 1Central Veterinary Research Laboratory (LaCIV), Facultad de Ciencias Veterinarias y Pecuarias, Universidad de Chile, Santiago 8820808, Chile; eduard.martinez@ug.uchile.cl (E.M.); macarena.pino@ug.uchile.cl (M.P.); nazla.vargas@ug.uchile.cl (N.V.); ale.ssoza@gmail.com (A.S.); 2Escuela de Medicina Veterinaria, Facultad de Ciencias de la Vida, Universidad Andrés Bello, Santiago 8370251, Chile; 3Laboratory of Immunogastroenterology, Gastroenterology Unit, Medicine Department, Hospital Clínico Universidad de Chile, Santiago 8380420, Chile; carollbeltranm@uchile.cl; 4Department of Animal Preventive Medicine, Facultad de Ciencias Veterinarias y Pecuarias, Universidad de Chile, Santiago 8820808, Chile; 5Programa de Magister en Ciencias Animales y Veterinarias, Facultad de Ciencias Veterinarias y Pecuarias, Universidad de Chile, Santiago 8820808, Chile; 6Instituto de Ciencias Básicas y Preclínicas “Victoria de Girón”, Universidad de Ciencias Médicas de la Habana, La Habana 11300, Cuba; oliverperezmartin@gmail.com; 7Hospital Veterinario MEDIVET, Centro de Diagnóstico Veterinario VETPOINT, Santiago 8320000, Chile; ismaelpereiraveterinario@gmail.com

**Keywords:** canine chronic inflammatory enteropathy, inflammatory bowel disease, microbiota, innate immune response, adaptive immune response

## Abstract

**Simple Summary:**

Canine chronic inflammatory enteropathy (CIE) is a chronic disease affecting the small or large intestine and, in some cases, the stomach of dogs. This gastrointestinal disorder is common and is characterized by recurrent vomiting, diarrhea, and weight loss in affected dogs. The pathogenesis of IBD is not completely understood. Similar to human IBD, potential disease factors include genetics, environmental exposures, and dysregulation of the microbiota and the immune response. Some important components of the innate and adaptive immune response involved in CIE pathogenesis have been described. However, the immunopathogenesis of the disease has not been fully elucidated. In this review, we summarized the literature associated with the different cell types and molecules involved in the immunopathogenesis of CIE, with the aim of advancing the search for biomarkers with possible diagnostic, prognostic, or therapeutic utility.

**Abstract:**

Canine chronic inflammatory enteropathy (CIE) is one of the most common chronic gastrointestinal diseases affecting dogs worldwide. Genetic and environmental factors, as well as intestinal microbiota and dysregulated host immune responses, participate in this multifactorial disease. Despite advances explaining the immunological and molecular mechanisms involved in CIE development, the exact pathogenesis is still unknown. This review compiles the latest reports and advances that describe the main molecular and cellular mechanisms of both the innate and adaptive immune responses involved in canine CIE pathogenesis. Future studies should focus research on the characterization of the immunopathogenesis of canine CIE in order to advance the establishment of biomarkers and molecular targets of diagnostic, prognostic, or therapeutic utility.

## 1. Introduction

Canine chronic inflammatory enteropathy (CIE) is a term used for gastrointestinal diseases present for 3 weeks or longer, after extraintestinal diseases and intestinal diseases of infectious origin or neoplastic conditions have been ruled out [1]. As in humans, a few years ago, the term IBD was used for canine CIE. However, nowadays there is consensus among experts that CIE is the most appropriate term [1,2,3], since, clinically, there are fundamental differences between dog and human diseases.

Human IBD includes at least two different chronic disorders characterized by different patterns of inflammation of the intestinal wall: Crohn’s disease (CD) and ulcerative colitis (UC) [4]. Instead, CIE in dogs is defined by the response to medical treatment: food-responsive enteropathy (FRE); antibiotic-responsive enteropathy (ARE); immunosuppressant-responsive enteropathy (IRE/SRE); and non-responsive enteropathy (NRE) [1]. Specifically for ARE, Cerquetella et al. recently provided evidence establishing that the empirical use of antibiotics in dogs with IBD can have detrimental effects. Thus, the authors suggested the use of antibacterials only after histopathological evaluation of gastrointestinal biopsies, when endoscopy is not possible, after other therapeutic trials have been unsuccessful, or when there is evidence of adherent-invasive bacteria [5].

Additionally, canine CIE is histopathologically classified according to the affected intestinal segment (stomach, small intestine, or large intestine) and the predominant type of cellular infiltrate (lymphocytic plasmacytic, eosinophilic, neutrophilic, or granulomatous). For example, when the small intestine is affected, lymphoplasmacytic or eosinophilic enteritis is dominant, while when the large intestine is affected, lymphoplasmacytic, eosinophilic, histiocytic, ulcerative, and regional granulomatous colitis has been identified [6]. In this regard, a simplified histopathologic scoring system has recently been proposed [7]. This score provides objective and descriptive information on the extent of mucosal inflammation in the gastrointestinal tract of dogs with CIE, demonstrating great diagnostic utility and important correlation with clinical findings in canine CIE [7].

Therefore, the classification of the disease is very different in dogs and humans. There is no Crohn’s-like disease in dogs, and canine histiocytic ulcerative colitis (HUC), an enteropathy previously compared with human enteropathies affecting the boxer breed, is different from human UC. HUC is considered to be caused by enteroinvasive *E. coli* and, therefore, does not belong to the canine idiopathic CIE [8].

The therapeutic approach also differs between humans and dogs. In human IBD, the treatment goals are defined by STRIDE-II, the latest update of Selecting Therapeutic Targets in Inflammatory Bowel Disease (STRIDE), initiative of the International Organization for the Study of Inflammatory Bowel Diseases (IOIBD) [9]. Medical therapy based on different types of immunosuppressant drugs is central to the management of the human condition and is aimed at controlling the inflammation and achieving mucosal remission [4]. In contrast, treatment in dogs is currently mainly aimed at clinical remission and many dogs do not require any treatment other than dietary modification (FRE). A further subset improves with antibiotic therapy, and a small proportion requires immunosuppressant treatment [10]. Several human patients with IBD have undergone surgery [11]. In contrast, surgery is not an indicated treatment for canine CIE [1].

The exact pathogenesis of canine CIE is still unknown and likely multifactorial, involving genetic and environmental factors, intestinal microbiota disarrangement (dysbiosis), and a dysregulated host immune response [4]. There are still significant knowledge gaps as to the role of innate and acquired immune response and microbiota in canine CIE pathogenesis. In order to identify potential novel biomarkers to determine the diagnosis, prognosis, or therapeutic approach, this review highlights some of the most relevant immunological findings of innate and acquired immune response concerning canine CIE.

## 2. Innate Immune Response

Local innate immune response is the first line of defense against commensal and opportunistic pathogens. However, its role in the pathogenesis of CIE has not been completely elucidated.

### 2.1. Intestinal Microbiota

Several studies have described differences in the gastrointestinal microbiota among dogs with various gastrointestinal diseases [12,13]. Thus, intestinal dysbiosis has been proposed as an important factor involved in CIE pathogenesis [14]. In addition, the use of a dysbiosis index (DI) to assess changes in the intestinal microbiota has been shown to be useful in evaluating microbial changes in fecal samples from dogs with CIE. [15].

Each intestinal segment has a specific microbiota; the colon and rectum contain the most diverse populations [14]. *Bacteroides*, *Clostridium*, *Lactobacillus*, *Bifidobacterium* spp., and *Enterobacteriaceae* are predominant genera. By 16S rRNA sequencing, the phyla *Firmicutes*, *Bacteroidetes*, and *Fusobacteria* have been identified as 95% of the total bacterial population, followed by *Proteobacteria* and *Actinobacteria* (1–5%) [16].

Phylogenetic studies have identified a decrease in the proportion of Clostridia and an increase in Proteobacteria in the duodenum of dogs with CIE [12]. Data concerning the genera present in the large intestine are limited. Suchodolski et al. (2012) concluded that dogs with active CIE present with a decrease in *Faecalibacterium* spp., which produces anti-inflammatory peptides in vitro, and *Fusobacterium* phyla. There are no differences in the *Proteobacteria* members [8].

### 2.2. Mucosal Epithelial Barrier

Gastrointestinal mucus of the intestinal epithelia is the first physical barrier to reduce the exposure to aggressors [17]. In the small intestine, mucus forms a single removable layer and, in the colon, a double layer. Here, mucins are the major barrier with the transmembrane and gel-forming mucins [18,19,20], which have direct immunological effects by binding to the numerous lectin-like proteins found in immune cells [21].

Homeostatic maintenance of the barrier is central to preventing the entry of bacteria and toxins from the lumen [22,23,24,25,26]. In dogs with CIE, it is possible that the breakdown of barrier integrity and the immunological tolerance against intestinal symbionts lead to deregulated inflammation and disease. In turn, this breakdown may also be amplified by CIE. Additionally, pathophysiological or environmental factors may induce loss of mucus barrier integrity [27,28].

Goblet cells, crucial for epithelial restitution, produce small peptides called trefoil factors (TFFs) which protect and repair the epithelial surfaces. The expression of TFFs is upregulated in human IBD [29,30]. In dogs with CIE, TFF1 expression is elevated in the duodenum, where TFF3 expression is down-regulated in the colon, suggesting they may contribute to the deterioration of the epithelial barrier [31].

Another epithelial barrier component is P-glycoprotein (P-gp), a membrane-bound efflux pump involved in the transport of a wide range of small molecules, whose abnormal expression is observed in dogs with lymphoplasmacytic enteritis (LPE). Some LPE patients have increased P-gp expression in the apical surface membrane of villus epithelial cells in the duodenum, jejunum, and/or ileum. In other patients, P-gp expression is decreased [32]. An upregulation in P-gp expression has been identified in lymphocytes from lamina propria after prednisolone treatment in dogs with CIE, which may be considered a predictor of response to therapy [33].

### 2.3. Innate Immune Cells and Their Derived Molecules

The barrier between blood and endothelial cells is tightly controlled physiologically. The barrier establishes the type and numbers of inflammatory cells that migrate to the interstitial space where dendritic cells (DCs), macrophages and mast cells in the lamina propria, and intraepithelial lymphocytes (IELs) monitor tissues, contributing to intestinal homeostasis [34,35].

#### 2.3.1. Integrins

In human IBD, when the barrier is disrupted, an uncontrolled transfer of inflammatory cells from the blood to the intestinal tissue occurs [36]. The extravasation is mediated mainly by integrins that bind their counterpart receptors on the endothelial cells. The molecules mediating normal endothelial–leukocyte interaction are the same as the molecules engaged in human IBD (α4β1 (VLA-4), α4β7, αDβ2, JAM-A, E-selectin, P-selectin, CD31, and CD99), although their expression levels are upregulated by inflammation [37,38,39]. There is not much information about specific integrins overexpressed in dogs with CIE. However, a reduced expression of the β-integrin CD11c has been described. This finding suggests that canine CIE may have an imbalance in the intestinal CD11c+ DCs. However, further studies are needed to determine whether CD11c could be a useful diagnostic biomarker for canine IBD [40].

#### 2.3.2. Cytokines

In canine CIE, IL-8 may stimulate transmigration of neutrophils to the mucosa and luminal contents to eliminate microbes during intestinal inflammation [28,37]. In a study on German shepherd dogs with CIE, the mRNA expression of many cytokines such as IL-2, IL-5, IL-12p40, interferon-γ (IFN-γ), tumor necrosis factor-α (TNF-α), and transforming growth factor-β1 (TGF-β1) was higher in diseased animals compared to controls [41]. However, Jergens et al. previously described through a meta-analysis that healthy dogs showed mRNA expression for most cytokines including IL-2, IL-4, IL-5, IL-10, IL-12, IFN-γ, TNF-α, and TGF-β. They determined that only IL-12 mRNA expression was increased consistently in small-intestinal CIE, whereas CIE lacked consistent patterns of expression [42]. Additionally, it remains unclear whether epithelial cell-derived cytokines such as IL-25, IL-33, and thymic stromal lymphopoietin (TSLP) contribute to the development of canine CIE [43].

#### 2.3.3. Metalloproteinases

Matrix metalloproteinases (MMPs) 2 and 9 are endopeptidases that play an important role in the turnover of extracellular matrix and cell migration and activate and degrade chemokines, cytokines, growth factors, and junction proteins [44]. MMP-2 is produced by stromal cells [45,46] and MMP-9 mainly by neutrophils, followed by eosinophils, monocytes/macrophages, lymphocytes, and epithelial cells [45,47,48,49,50]. Both MMPs could be involved in the pathogenesis of canine CIE. Although their role in this pathology has not been completely elucidated, they are upregulated in dogs with CIE [51].

#### 2.3.4. Neutrophils

In the duodenal mucosa of dogs with CIE, an increase in neutrophils is associated with disease severity [48]. Serum perinuclear anti-neutrophilic cytoplasmatic autoantibodies (pANCA) [52] and blood neutrophil-to-lymphocyte ratio (NLR) [53] have been proposed as biomarkers of canine CIE severity. NLR has also been proposed as a useful marker to differentiate FRE from IRE, with clinical utility to subclassify the canine CIE. However, it is important to note that NLR may not be useful for NRE subclassification [54].

In addition, calgranulin-C, a protein secreted by activated neutrophils and monocytes/macrophages, and myeloperoxidase (MPO) activities increase in the mucosa of the duodenum and colon of dogs with CIE, and MPO also increases in the ileum and cecum. However, none have been related to the clinical outcome of patients [55].

Calprotectin, another protein released by activated mononuclear cells, has increased expression in canine intestinal mucosa [56] and is used as a diagnostic and prognostic factor in human IBD [57,58]. Determination of fecal calprotectin concentration is a useful screening test for human IBD diagnosis, reducing the need for colonoscopy by 66.7% [59]. Serum calprotectin concentrations may also be a useful biomarker for the detection of inflammation in dogs, but the use of certain drugs such as glucocorticoids could limit clinical usefulness [60]. The authors also showed that fecal calprotectin could be used as a possible marker for assessing the severity of gastrointestinal inflammation in dogs with CIE [61]. Additionally, a recent meta-analysis concluded that fecal calprotectin concentration is one of the most promising biomarkers of gastrointestinal functionality in dogs [62].

#### 2.3.5. Macrophages

Macrophages participate in the host defense against infections and also remove apoptotic cells and remodel the extracellular matrix [63]. These cells are differentiated into two subtypes, termed M1 and M2. M1 macrophages initiate and maintain inflammatory processes, whereas M2 are associated with the resolution of chronic inflammation and the promotion of tissue repair [64].

Macrophages are present in large amounts in the intestine, primarily the colon, which has a high bacterial load. Intriguingly, these macrophages release mediators that promote homeostasis and thereby do not contribute to a proinflammatory environment [65,66,67,68]. This selective inertia is important to maintain the homeostasis and epithelial integrity. Disturbances in this condition may be involved in the pathogenesis of UC and CD in humans [69,70,71].

An IBD murine model determined that under healthy conditions, macrophages display an anti-inflammatory phenotype (M2) with expression of MHC II, CD163, and IL-10 production. Under pathological conditions, monocytes differentiate into pro-inflammatory macrophages (M1), characterized by the expression of inducible NO synthase (iNOs), CD64^+^HLA-DR^hi^ CD14^lo^, producing pro-inflammatory cytokines and chemokines such as TNFα, IL-1β, IL-6, IL-12, IL-23, and CCL11 [69,72].

Increased numbers of macrophages have also been identified in the duodenal mucosa of dogs with CIE [73]. Similar to humans with UC, boxer breed dogs with histiocytic ulcerative colitis (HUC) have higher infiltration of periodic acid-Schiff (PAS)-positive macrophages in the lamina propria in colonic and non-colonic affected regions, with a decrease in Goblet cells and an increase in MHC class II expression in enterocytes [74]. However, a later study revealed that HUC in boxer dogs is caused by enteroinvasive *E. coli* and can be successfully treated with fluoroquinolones such as enrofloxacin. Therefore, canine HUC is considered rather an infectious disease than belonging to the canine idiopathic CIE complex [8,75]. Similarly, granulomatous colitis in young French bulldogs has been also associated with the presence of invasive *E. coli* [75].

A recent study on dogs of different breeds and with or without CIE determined a reduced number of total macrophages but a slightly increased number of CD64^+^ macrophages, contributing to CIE pathogenesis [76]. Another study characterizing macrophages in the duodenum by immunohistochemistry, evaluated calprotectin (CAL) as a marker of early differentiated macrophages (M1) and CD163 expression as a marker of duodenal resident macrophages (M2), before and after treatment. This study demonstrated that macrophages play an important role in dogs with CIE. In particular, in dogs with FRE and IRE, the CD163^+^/CAL ratio is lower than in healthy dogs at diagnosis, and it is normalized after treatment in dogs with FRE. No significant differences were observed in dogs with ARE [77].

A study analyzing transcription nuclear factor (NF-κB) activation during mucosal inflammation in situ in dogs with CIE, identified significantly more macrophages/mm^2^ with increased activity of the NF-κB pathway in the lamina propria [78], suggesting a role of NF-κB and derived pro-inflammatory cytokines in CIE.

#### 2.3.6. Eosinophils

Eosinophils are granulated cells that contribute to the host defense against parasites and play an important role in local immune regulation. In humans with IBD, eosinophils are increased in number along with IL-5 production, prompting circulation and activation [79,80]. Eosinophils infiltrating the intestinal mucosa release granules composed of crystalloid-containing core encapsulating cationic proteins and leukotriene C4 [80,81,82].

Canine CIE classification is based on the predominant type of inflammatory cells. The second most diagnosed form of CIE is in the small intestine where lymphoplasmacytic and eosinophilic enteritis are observed [83,84,85]. A study demonstrated that dogs with CIE have a significantly higher number of degranulated eosinophils in the lower region of the lamina propria, while the upper region has a significantly higher number of degranulated and intact eosinophils [6]. Comparing human and canine eosinophilic gastroenteritis showed that both pathologies have clinical and histopathological similarities [85], and canine eosinophilic gastroenteritis could be a good model for its human counterpart.

Various noninvasive tools and markers have been studied to identify eosinophil activation in the GI tract, including peripheral eosinophil counts and serum 3-bromotyrosine concentrations (3-BrY) [86]. Serum 3-BrY concentrations were also higher in dogs with SRE/IRE than in those with FRE or healthy control dogs. Thus, 3-BrY may serve as a noninvasive biomarker for CIE diagnosis and prognosis [87].

Another potential marker of eosinophil activity is the soluble epoxide hydrolase (sEH), a molecule with a proinflammatory role by metabolizing anti-inflammatory epoxyeicosatrienoic acid to proinflammatory diols. In a murine model, the use of a specific inhibitor of sEH significantly inhibited eosinophil migration, suggesting that sEH plays an important role in the migration of eosinophils to the gastrointestinal system [88].

Interestingly, a study of 30 dogs with CIE found that a significant number of dogs with CIE showed severe (n = 8) or moderate and mixed eosinophilic inflammation (n = 12). Future studies should be performed to further characterize the role of these eosinophils in canine CIE [89].

#### 2.3.7. Mast Cells

Mast cells, involved in the immediate and delayed defense against foreign antigens, also release mediators that affect the mucosal barrier [90]. More recently, a possible relationship between mast cells and host microbiota in human IBD pathogenesis has been proposed. This interaction is crucial to prevent mast cell hyper-reactivity. However, when microbiota genera are expanded, the interaction increases, favoring permeability and release of immunomodulatory molecules that promote inflammation [91].

In dogs with CIE, an increase in mast cells in the area of the eosinophilic gastroenterocolitis has been described, suggesting a role of type I hypersensitivity [85]. Thus, dogs with CIE have significantly more cells positive for IgE protein and mast cells in the mucosa, but their main location is mesenteric lymph nodes [92]. Moreover, a study defining the distribution and types of mast cells in the normal gastrointestinal tract of canines detected fewer mast cells in the villus area compared to the crypt areas; tryptase-positive mast cells (MC_T_) were the most abundant cell type, followed by chymase- and a few tryptase- and chymase-positive mast cells (MC_C_, MCT_C_) [93]. However, in dogs with lymphocytic-plasmacytic or eosinophilic gastroenterocolitis, there was a decrease in the number of metachromatically stained granule-containing mast cells and a decrease in the number of the three types of mast cells identified (MC_T, -C, -TC_), suggesting a mast cell degranulation or a Th1 predominant pattern [94].

N-Methylhistamine (NMH) is a stable metabolite of histamine and may be used as a marker of mast cell degranulation and gastrointestinal inflammation [95]. A study by Berghoff et al. showed that some dogs with CIE have increased fecal and/or urinary NMH concentrations, which could indicate increased mast cell activity. However, they were unable to definitively demonstrate such an association. In the same study, the authors suggest that urinary NMH concentrations could have clinical utility as a biomarker of chronic gastrointestinal inflammation, but this area remains to be explored further [96].

#### 2.3.8. Natural Killer Lymphocytes and Natural Killer Cells

Natural killer lymphocyte (NKT) and natural killer (NK) cells also have a role in human IBD. Th2 cytokines such as IL-13, IL-5, and IL-4, involved in UC, are partly produced by NKTs [97,98,99,100]. However, NKTs have a dual role as they play a protective role in a dextran sulfate sodium-induced colitis model [101] and a detrimental role in an oxazolone-induced model [102]. An increase in the cytotoxic CD56^+^CD16^+^ NK cell subset in the lamina propria in human IBD patients [103] and a decrease in the NKp44^+^/NKp46^+^ ratio in biopsies of CD patients have been demonstrated [104]. Recently, a meta-analysis evaluating the role of killer-cell immunoglobulin-like receptor (*KIR*) genes of IBD susceptibility in humans found that *2DL5* and *2DLS1* genes are associated with an increased risk of UC, while the *2DS3* gene is associated with a decreased risk of CD development [105]. Additionally, experimental treatment with monoclonal antibodies against NK Group 2D (NKG2D), a constitutively expressed receptor whose ligand is highly expressed in human IBD, has resulted in remission of CD in some patients [106].

The role of NK and NKT cells in CIE development in dogs has not been studied. Due to the relevance in human IBD, further investigation in dogs could provide relevant information.

#### 2.3.9. Natural Antibodies

Natural antibodies (Nabs), IgM and other pre-existent classes of immunoglobulins circling in plasma, are essential components of innate immunity reacting against foreign antigens and microbe-derived substances and activating the classical pathway of complement activation [107,108]. Its role in canine CIE has not been established but should be considered because, in a murine model, homeostatic intestinal IgAs are natural polyreactive antibodies with innate specificity to microbiota [109].

#### 2.3.10. S100/Calgranulins and RAGE Receptors

S100/calgranulins are a group of three phagocyte-specific damage-associated molecular pattern molecules (DAMPs) [110,111] that include S100A12 (calgranulin C) and the S100A8/A9 (calprotectin or calgranulin A/B) complex. The proteins are produced by activated macrophages and neutrophils and accumulate at sites of inflammation [3]. Recently, it has been suggested that fecal S100A12 and fecal calprotectin concentrations are clinically useful markers of gastrointestinal inflammation in dogs [3]. Fecal canine S100A12 concentrations are increased in dogs with CIE, associated with clinical disease activity, the severity of endoscopic lesions, and the severity of colonic inflammation in dogs with CIE [112,113].

Additionally, a recent study evaluated the expression of gastrointestinal mucosal receptor for advanced glycation end products (RAGE), which are considered molecular pattern receptors with relevance to inflammation in dogs with CIE, and its binding to canine S100/calgranulin ligands. CIE in dogs is associated with decreased serum sRAGE concentrations [114] and an increase in epithelial RAGE expression in the duodenum and colon [113] suggesting a dysregulated sRAGE/RAGE axis [114]. The epithelial RAGE expression in the duodenum and colon was significantly higher in dogs with CIE than in healthy controls, with a pattern of overexpression in the ileum and underexpression in the stomach. Thus, although the role of this axis in canine CIE is not completely understood, this axis might be a possible therapeutic target for dogs with CIE, with utility as a therapeutic model for humans [115]. 

#### 2.3.11. Pattern Recognition Receptors (PRRs): Toll-Like Receptor and NOD-Like Receptors

Both commensals and pathogenic bacteria express pathogen-associated molecular patterns (PAMPs) on their surface, which are recognized by host pattern recognition receptors PRRs [116]. Among the best characterized PRRs are Toll-like receptors (TLRs) and NOD-like receptors (NOD) [117,118]. Under normal conditions, PRRs recognize antigens from food and commensal bacteria inducing tolerogenic responses. In canine CIE, these antigens, which normally induce immune tolerance, trigger an inflammatory response, with proliferation of T lymphocytes and production of several pro-inflammatory cytokines [116]. Thus, an increased TLR2, TLR4, and TLR9 mRNA expression in dogs with CIE has been described [119,120].

In human IBD, some mutations in these PRRs have been associated with its development [121]. Similarly, several mutations in PRRs have been also associated with the development of canine CIE. Single-nucleotide polymorphisms (SNPs) associated with TLR4 and TLR5 [122] and NOD2 [123] have been identified in German shepherd dogs with CIE. Additionally, a genetic component has been established in canine CIE, with a predisposition of certain breeds. Among those predisposed breeds are German shepherd dogs, Weimaraners, Rottweilers, border collies, and boxers [124].

Subsequently, a TLR5 haplotype has been identified, which is associated with a hypersensitivity to flagellin, exacerbating inflammatory pathways in dogs carrying this haplotype, increasing the risk of developing CIE [125].

While all these results provide valuable information for the development of possible genetic markers of CIE, it should be considered that CIE is a polygenetic disorder. Furthermore, these potential genetic markers would in many cases be expected to be breed-dependent [116].

## 3. Adaptive Immunity

The intestinal adaptive immune response is composed of CD4^+^ T cells, IgA-producing B cells in Peyer’s patches (PPs) and lamina propria, and intestinal epithelial lymphocytes (IELs), which play a critical role in maintaining immune tolerance [126,127].

### 3.1. T Helper CD4^+^ Lymphocytes

In homeostasis, intestinal professional antigen-presenting cells (APCs) migrate to mesenteric lymph nodes where they present antigens and activate CD4^+^ T lymphocytes mediating pathogenic immunity or mucosal tolerance and barrier integrity contributing to the expansion of T helper cells or regulatory T-cells (Treg), respectively. Alteration of T cells leads to an imbalance between regulatory and effector cells, resulting in tissue inflammation, where pro-inflammatory cytokines such as TNF-α, IL-1, IL-6, IL-8, IL-12, IL-18, IL-23, and chemokines are secreted [28,128].

T helper cells (CD4^+^) are a subpopulation of T lymphocytes. The main T helper subpopulations are Th1, Th2, Th17, and Treg. The differentiation of T helper lymphocytes varies according to the type of antigen that the APC faces and the length of exposure. These cells are essential components of adaptive immunity since they secrete specific cytokines in response to MHC-II-dependent peptide recognition and co-stimulatory signals from APCs [28,129].

Naive helper T-lymphocyte (Th0) differentiation into Th1 occurs when DCs or professional phagocytes primarily release IL-12 [128]. These Th1 cells induce cell-mediated effector responses, such as cytotoxicity and immunity to intracellular organisms, secreting IL-2, IL-12, INF-γ, and TNF-α [130]. The transcription factors associated with their differentiation are STAT4 and T-bet [129,130].

When activated DCs secrete IL-4, Th0 cells differentiate into Th2, with the participation of transcription factors GATA-3 and STAT6 [129]. Th2 cells produce IL-6, IL-4, IL-5, and IL-13, activating B lymphocytes to produce IgE and recruit eosinophils. This Th2 response is mainly associated with helminth infections and allergies [129,130].

If activated DCs mainly secrete IL-6 and TGF-β, Th0 differentiate to Th17. Th17 cells are characterized by the transcription factor RORγt and production of IL-17A, IL-17F, and IL-22 cytokines, which subsequently trigger inflammatory signaling cascades and lead to the recruitment of innate immune cells [129,131].

Finally, Th0 differentiate to Treg cells in the presence of transcription factor Foxp3. Treg cells secrete cytokines that have anti-inflammatory effects such as IL-10 and TGF-β [132].

In humans, an association between the predominant immune profile and the type of IBD induced has been established. CD is associated with a Th1/Th17 response, whereas UC has a predominant Th2/Th9 response [133,134,135].

In dogs, a predominant profile for CIE has not been determined [130]. A meta-analysis of intestinal cytokine mRNA expression showed a balance in the expression of pro-inflammatory and anti-inflammatory cytokines in German shepherd dogs with CIE [42]. Previously, an increase in the expression of IL-2 and TNF-α in dogs with colitis was observed [136]. A more recent study evaluated IL-25, IL-33, and TSLP mRNA expression in the intestinal epithelial cells (IECs) of the duodenal and colonic mucosa of dogs with FRE [43]. These three cytokines enhance Th2-dominated immunity by stimulating DCs, innate lymphoid cells, basophils, and mast cells [137]. This study showed that IL-33 mRNA expression was significantly lower in the duodenum of dogs with FRE than in healthy dogs. These results suggest that a Th2-response is not induced in canine CIE. However, further studies are needed to understand the role of IL-33 in canine CIE [43].

A Th17 response is involved in the pathogenesis of IBD in humans [138]. Although the role of these cells has not been completely described in canine CIE [139,140], an increase in the expression of IL-17A, IL-23p19, and Il-12p35 has been identified [141,142].

In addition, a low number of Treg cells and mRNA expression of IL-10 and TGF-β has been described in dogs with lymphoplasmacytic enteritis. Since TGF-β is essential for Th0 differentiation into Treg, a decrease in TGF-β expression may contribute to a decrease in Treg cells in the duodenum in dogs with CIE [143,144].

There are discrepancies associated with predominant Th profiles in canine CIE that could be explained by differences in the inclusion and exclusion criteria and the clinical heterogenicity of dogs in different studies [28,145].

Signal transducer and activator of transcription 3 (STAT3), an essential transcription factor for the differentiation of Th17 lymphocytes, plays an important role in the pathogenesis of human IBD [146]. For its activation, STAT3 is phosphorylated (pSTAT3), contributing to the intestinal homeostasis and intestinal wound healing, and stimulating the release of several anti-inflammatory cytokines [147,148].

A recent study found significant activation of STAT3 in the duodenal mucosa of dogs with different subtypes of CIE. Higher expression was found in the epithelium and lamina propria of the crypt area in the FRE group than in the PLE group, where pSTAT3 upregulation was more dominant in the epithelium of the crypt and villus area. Only the SRE group featured pSTAT3 upregulation in both areas compared to the control group. Thus, pSTAT3 upregulation has been proposed as a characteristic of SRE and an important clinical marker for active mucosal inflammation in CIE [149].

### 3.2. Intestinal Intraepithelial Lymphocytes

Intestinal intraepithelial lymphocytes (IELs) are an important cell population at mucosal sites. Two major subtypes of IELs have been described, involving both pro- and anti-inflammatory functions [126]. These subtypes correspond to the conventional IELs, characterized by the expression of the T-cell receptor (TCR) αβ^+^ with co-receptor clusters CD4^+^ and CD8^+^ and a non-conventional IEL expressing TCRαβ^+^ or TCRγδ^+^ combined with co-receptor CD8αα^+^ [150,151]. Human patients with CD show higher levels of TCRγδ^+^ T cells in the inflamed colonic mucosa [152]. In dogs, IELs are diffusely scattered throughout the small-intestinal villus epithelium [93], but an increase in the number of TCRγδ^+^ T cells has been observed in dogs with CIE [153].

### 3.3. B-Lymphocytes

B cells are an essential component of mucosal immunity with an important role as APCs, mainly in the secondary immune response, modulating the microbiota diversity, maintaining the integrity of the intestinal barrier [154,155], and secreting a variety of antibodies. IgA in the lamina propria neutralizes luminal microbes transporting them from the mucosal epithelium to the lumen, among other functions [155]. In humans, an increase in plasma and serum cell infiltration and local IgA, IgM, and IgG in the intestinal mucosa has been described in patients with CD [156,157] and UC [158,159,160]. Similarly, dogs with CIE show an increase in the number of B lymphocytes in the bloodstream [161] and intestinal mucosa [162] and an increase in IgG^+^, IgG3^+^, and IgG4^+^ in plasma cells [73].

In the intestinal mucosa, dimeric IgA is secreted and transported through the epithelium into the intestinal lumen. IgA is important in mucosal defense through (1) preventing the passage of pathogens, (2) influencing mucosal defense mechanisms by preventing the spread of pathogens into intestinal tissue, and (3) preventing infections and dysbiosis [163,164].

Canine CIE has been associated with a reduction in IgA levels in intestinal mucosa and feces [144] and is associated with a decreased pattern of expression of the transmembrane activator and calcium-modulating cyclophilin–ligand interactor (TACI) and a decreased B cell-activating factor of the TNF family (BAFF-R). In addition, a hypermethylation of TNFRSF13B and TNFRSF13C loci has been observed, possibly associated with a defect in IgA class switching [165,166]. All this suggests a role of mucosal IgA deficit in CIE [28].

A recent study showed that dogs with CIE have higher levels of IgA specific to serological markers such as polynuclear leukocytes, bacterial OmpC, calprotectin, gliadins, and bacterial flagellins. However, further studies demonstrating its clinical relevance are required [167].

Interestingly, Soontararak et al. [168] showed that bacteria present in the gut of dogs with CIE had significantly higher levels of IgG fixation. Furthermore, these IgG levels appeared to be directed against certain dysbiotic bacteria, mainly Actinobacteria. These IgG-coated bacteria induce higher production of TNF-α by macrophages, indicating a pro-inflammatory effect in dogs with CIE, similar to that observed in humans [168].

## 4. Cross-Talk in the Immune Responses and Their Possible Role in the Pathogenesis of Chronic Inflammatory Enteropathy (CIE) in Dogs

The immune system is traditionally classified into the innate and adaptive immune response. However, both responses are part of a dynamic process, where molecules and cells from both innate and acquired responses are strongly integrated. Thus, cells of the innate immune response recognize pathogens and tissue damage triggering an inflammatory response, and DCs, T lymphocytes, and B lymphocytes drive an acquired immune response, which is concomitantly induced [169].

In CIE, barrier integrity breakdown leads to an exaggerated immune response and loss of immunological tolerance, facilitating microbiota influx and phagocytosis [27,28] initiating the innate immune response. This response modulates the expression of different molecules and cytokines contributing to recruiting and activating different types of inflammatory cells [33,40,41,42,43,72,73,76,89,92,93,94,170,171]. As a consequence, antigen-presenting cells (APCs), such as local DCs or macrophages, and phagocyte pathogens migrate to mesenteric lymph nodes to present antigens and activate different subpopulations of T lymphocytes. Although in dogs, a predominant profile of lymphocytes for CIE has not been determined [130], both pro-inflammatory and anti-inflammatory cytokines have been identified [42,43,136,137]. This difference from human IBD could be explained by the complex classification of CIE and other factors involved in its pathogenesis which may interfere with and modify the immune response.

The main findings related to the role of the innate and adaptive immune response in the pathogenesis of canine CIE and other enteropathies are summarized in Figure 1 and Table 1.

## 5. Concluding Remarks

Despite advances in the complex molecular mechanisms explaining CIE in dogs, the contribution of the immune response in the pathogenesis of CIE is not completely understood. To determine the role of barrier integrity breakdown and the loss of immunological tolerance against intestinal symbionts, the microbiota–immune-system interaction is essential to completely understand canine CIE pathogenesis and modulate the clinical consequences. There are important gaps in the knowledge about the role of some molecular and cellular components forming part of the innate immune response which has been identified as an experimental therapeutic target in human IBD. The adaptive immune response also plays a critical role in maintaining immune tolerance toward symbiotic bacteria, integrity of the intestine barrier, and gut homeostasis. In human IBD, CDs-Th1/Th17 and UC-Th2/Th9 associations have been described. However, in canine CIE, a predominant immune profile has not been clearly established due to the heterogenicity in the inclusion/exclusion criteria and clinical aspects of patients enrolled in different studies.

In this review, we characterized different cells and molecules which have been identified as playing a role in the immunopathogenesis of canine CIE. Future research should advance the characterization of canine CIE immunopathogenesis in order to identify future biomarkers and molecular targets of diagnostic, prognostic, and potential therapeutic utility.

## Figures and Tables

**Figure 1 animals-12-01645-f001:**
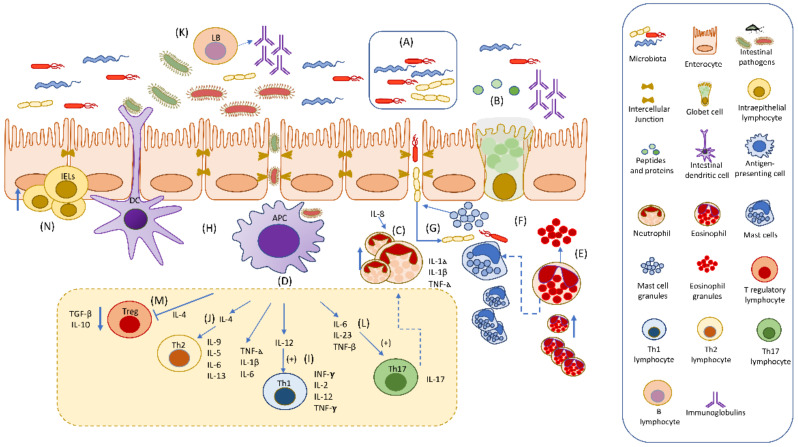
Cross-talk in the immune responses and their possible role in the pathogenesis of chronic inflammatory enteropathy (CIE) in dogs. (**A**) In CIE, barrier integrity breakdown leads to pathological inflammation and loss of immunological tolerance. A decrease in bacteria forming part of the physiologic microbiota such as Faecalibacterium spp. and Fusobacterium phyla has been identified. (**B**) Goblet-cell-derived peptide and glycoprotein expression from enterocytes is up- or down-regulated. (**C**) During intestinal inflammation, IL-8 contributes to neutrophil recruitment and secretion of IL-1α, IL-1β, and TNF-α. (**D**) Phagocytic cells, including antigen-presenting cells (APCs) such as dendritic cells (DCs) and macrophages phagocyte microorganisms and differentiate to a pro-inflammatory phenotype, secreting IL-12, TNF-α, IL-1β, and IL-6. Additionally, (**E**) the number and degranulation of eosinophils and (**F**) the number and granules of mast cells are increased. (**G**) When the barrier is disrupted, mucosal permeability facilitates microbiota influx and phagocytosis. The microbiota influx also promotes microbiota–mast-cell interaction and releasing of immunomodulatory molecules. (**H**) APCs phagocyte pathogens and migrate to mesenteric lymph nodes to present antigens and activate different subpopulations of CD4+ T lymphocytes. (**I**) Th1 is stimulated by IL-12 released by APC and secretes mainly IL-2, IL-12, INF-γ, and TNF-α, while (**J**) Th2 is stimulated by IL-4 released by APC to produce IL-6, IL-4, IL-5, IL-9, and IL-13. (**K**) There is an increase in B lymphocytes and in IgG, IgG3, and IgG4 levels but a decrease in IgA in the intestinal mucosa. (**L**) When activated, APCs secrete IL-6 and TGF-β, and a Th17 subpopulation is activated. This population secretes IL-17A, IL-17F, and IL-22, which participate in neutrophil recruitment. Finally, (**M**) Treg subpopulations secrete IL-10 and TGF-β, contributing to control of the immune response. However, in dogs, a Th predominant immune profile for CIE has not been completely determined. In addition, (**N**) an increase in intestinal intraepithelial lymphocytes has been identified in dogs with CIE.

**Table 1 animals-12-01645-t001:** The innate and adaptive immune response in dogs with CIE.

Innate Immune Response	References
*Microbiota*	
A decrease in proportion of Clostridia and increase in proportion of Proteobacteria in the duodenum	[12]
A decrease in Faecalibacterium spp and Fusobacteria	[12]
*Mucosal epithelial barrier*	
Pathophysiological or environmental factors could induce loss of the mucosal barrier integrity and immune tolerance against intestinal symbionts	[27,28]
Trefoil factor (TFF) 1 expression is elevated in the duodenum, whereas TFF3 expression is down-regulated in the colon, suggesting that it contributes to impaired epithelial barrier function	[31]
Abnormal P-glycoprotein (P-gp) expression is observed in dogs with lymphoplasmacytic enteritis (LPE)	[32]
Upregulation of P-gp expression in lamina propria lymphocytes after prednisolone treatment	[33]
*Innate immune cells and derived molecules*	
A reduced expression of the β-integrin CD11c	[40]
An increase in neutrophils as a factor associated with severity	[73]
Perinuclear anti-neutrophil cytoplasmic autoantibodies (pANCA) and neutrophil-to-lymphocyte ratio (NLR) as biomarkers of severity	[52,53]
Calgranulin-C and myeloperoxidase (MPO) activities are increased in the duodenum and colon of dogs with chronic enteropathies, and myeloperoxidase (MPO) is also increased in the ileum and cecum. Calprotectin is overexpressed and released by activated mononuclear cells in canine CIE	[55,56]
Matrix metalloproteinases (MMPs)-2 and -9 are upregulated in dogs with CIE	[51]
Increased numbers of macrophages in the duodenal mucosa	[73]
An increase in macrophage infiltration in the lamina propria in colonic and noncolonic affected regions, a decrease in Goblet cells, and an increase in MHC class II expression in enterocytes of boxer breed dogs with CIE	[74]
An increase in macrophages/mm^2^ with increased NF-κB pathway activity in the lamina propria	[78]
Degranulated eosinophils in the lower region of the lamina propria and degranulated and intact eosinophils in the upper	[6]
Increased concentration of Serum 3-BrY (associated with eosinophil activation) in dogs with SRE/IRE compared to those with FRE or healthy control dogs	[87]
Increased mast cells in the area of eosinophilic gastroenterocolitis	[85]
More IgE-positive cells and mast cells in the mucosa and mesenteric lymph nodes	[92]
A decrease in metachromatically stained granules and mast cells in dogs with lymphocytic-plasmacytic or eosinophilic gastroenterocolitis	[94]
Increased fecal and/or urinary NMH concentrations in some dogs with CIE	[96]
Increased fecal S100A12 concentrations associated with clinical disease activity, the severity of endoscopic lesions, and the severity of colonic inflammation	[112]
Decreased serum sRAGE concentrations in canine CIE	[114]
Overexpression of epithelial RAGE along the gastrointestinal tract in dogs with CIE	[115]
**Adaptive Immune Response**	
*T helper lymphocytes (CD4+)*	
A balance in the expression of proinflammatory and anti-inflammatory cytokines in German shepherd dogs	[42]
An increase in IL-2 and TNF-α expression in dogs with colitis	[136]
An increase in IL12p40-associated mRNA in dogs with lymphocytic-plasmocytic enteritis and lymphocytic-plasmocytic colitis, when the duodenum is affected. An increase in IL-4 mRNA expression when the colon is affected	[42]
An increased expression of IL-17A, IL-23p19, and Il-12p35	[139,140,141,142]
Low number of Treg cell and IL-10 and TGF-β mRNA expression in dogs with lymphocytic-plasmocytic enteritis	[143,144]
*Intestinal intraepithelial T lymphocytes*	
Increased numbers of TCRγδ+ cells	[93,153]
*B lymphocytes*	
Increased numbers of B lymphocytes in the bloodstream and intestinal mucosa. IgG+, IgG3+, and IgG4+ also increase in plasma cells	[73,161,162]
Reduced IgA levels in intestinal mucosa, feces, and peripheral blood	[28,144]
High levels of specific IgA against serological markers such as polynuclear leukocytes, bacterial OmpC, calprotectin, gliadins, and bacterial flagellins	[167]
An increase in IgG-coated gut bacteria, which induce increased production of TNF-α by macrophages	[168]

## Data Availability

Not applicable.

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
