# Peer review of "Elucidating the Role of Innate and Adaptive Immune Responses in the Pathogenesis of Canine Chronic Inflammatory Enteropathy—A Search for Potential Biomarkers"

_animals, 2022, doi:10.3390/ani12131645_

Round 1
Reviewer 1 Report
Dear authors,
I consider your work an important contribution to the study of canine IBD.
Simple summary
Line 25 - Please delete "the presence of microorganisms" as the issue more than the presence of some specific microorganism is dysbiosis ("dysregulation of the microbiota"). Remember that IBD is idiopatic and not related with any particular pathogenic microorganism.
Keywords: Chronic enteropathies in dogs; Inflammatory Bowel Disease; microbiota; innate immune response; adaptive immune response.
2.1. Microbiota
Lines 83 to 87 - Suchodolski et al. concluded that dogs with active IBD present a decrease in Faecalibacterium spp., which produces anti-inflammatory peptides in vitro, and Fusobacterium phyla, but there are no differences in the Proteobacteria members [8, 9].
Figure 1 as a whole should be referred to earlier in the text or alternatively move figure 1 to a more advanced position in the text, after all its points have been cited. At present some parts are referred to before the appearance of the figure and others after.
2.3.1. Integrins
Line 169 - In human IBD, when the
2.3.3. Neutrophils
Line 200 - as a prognostic factor in human IBD [56,57].
I think the occasion should be taken to talk about the importance of this fecal marker of inflammation of the intestine.
3.1. T helper O CD4+ lymphocytes
Line 342 - German et al. [75]
Table 1
Regarding table 1, reference 8 refers to the dog and 9 to the man.
Conclusions
of human IBD.
Author Response
Dear Reviewer,
Thank you very much for your comments and corrections. We have worked to incorporate all of your recommendations. In addition, a native speaker has corrected the English throughout the manuscript.
Our responses to your review are detailed in the attached document.
Kind regards,
Daniela Siel.

Reviewer 2 Report
This paper is very confusing and very incomplete according to me. While the goal is canine IBD, most of the quotes are about human studies and are irrelevant.
I do think the name has moved from IBD to chronic enteropathy. I would recommend that this be changed through the article and the title. Dandrieux JRS. Inflammatory bowel disease versus chronic enteropathy in dogs: are they one and the same? J Small Anim Pract. 2016;57(11):589–599.
develops (pathogenesis) – pick one line 24 - this is redundant
Unlike the human IBD, canine IBD is classified 37 on basis of the intestinal segment that are affected and the predominant type of inflammatory cells. 38 -What about food-responsive enteropathy (FRE), antibiotic-responsive enteropathy (ARE), and immunosuppressant-responsive enteropathy (IRE) or SRE?
Clinically, IBD is characterized by abdominal pain, diarrhea, bloody stool, weight 55 loss, and inflammation – I disagree these are not the typical signs of CE
How is CE labeled by affected intestinal segment, what about the stomach?
Since domestic dogs are living in indoor environments and feeding 74 commercial or natural diets, the gut microbiota has been modified and complexed. Where is this coming from? Reference?
Suchodolski, Markel, Garcia-Mazcorro, Unterer, 83 Heilmann, Dowd, Kachroo, Ivanov, Minamoto, Dillman, Steiner, Cook and Toresson [8] I would just say Suchodolski et al.
Other important component is the complement system (C). Several C components are 149 secreted by enterocytes [28]. In humans, the expression of C1 [29,30], C4 [31-33], C3 [33] 150 and factor B [31,32] have been detected in different intestinal segments. Reports have 151 demonstrated C deposition in ulcerative colitis (UC) in human patients [34] and a role of 152 C component and its regulatory proteins, from alternative and classical pathways, in 153 Crohn’s disease (CD) and UC inflammatory damages, respectively [28]. There is no 154 information about the role of C in the pathogenesis of IBD in dogs. 155 Additionally, a pathological angiogenesis has been observed in human IBD, causing an 156 enlargement of the superficial area where leukocytes encounter endothelial cells, 157 impacting its interaction in the intestinal mucosa [35-38]. In dogs with IBD, its role has not 158 been studied. -This is not a comparative paper. This is irrelevant information.
In dogs with IBD, an increase in neutrophils has been 184 reported as a factor associated to severity [48] (Figure 1C). 185 - Why is this in this section?
‘In local wound sites, neutrophils produce vascular endothelial growth factor and pro- 188 resolving lipid mediators derived from omega-3 fatty acids, as well as arachidonic acid 189 metabolites, lipoxin A4, protectin D1, and resolvin E1 [49,50]. Protectin D1 and resolving 190 E1 decrease neutrophil recruitment and increase macrophage phagocytosis of apoptotic 191 neutrophils [51]. I’ relevance?
What about Intestinal mucosal S100A12 ?
What about studies with 3-bromotyrosine and Eosinophilic IBD?
What about NMH with and mast cells?
Nothing about IgA, Systemic levels of the anti-inflammatory decoy receptor soluble RAGE (receptor for advanced glycation end products) etc?
Author Response

(The authors gave the same response as above.)

Reviewer 3 Report
The authors gave a comprehensive summary of the current knowledge about innate and adaptive immune response in canine spontaneous IBD and compared it to human IBD and experimental models of human IBD. Such a summary is currently missing in literature and I want to thank the authors for compiling the results of several world wide acting research groups. However, there are some issues that need to be addressed before considering publication.
Comments for major revision
- The introduction is missing a clear definition of canine IBD as it has been given e.g., by Dandrieux 2016: ""When the term inflammatory bowel disease is used for dogs, it typically implies that treatment trials with diet and subsequently antibiotics have failed, inflammation has been demonstrated and an immunosuppressant will be needed." See: https://pubmed.ncbi.nlm.nih.gov/27747868/. This is so far important that canine IBD differs substantially from human IBD. There is no Crohn's like disease in dogs. Canine Histiocytic Ulcerative Colitis (HUC) has been proven to be caused by enteroinvasive E. coli and does therefore not belong to the canine idiopathic IBD responding to immunosuppressive treatments. Using e.g. corticosteroids in Boxers with HUC leads to worsening of clinical signs and even dead. (See Mansfield et al. 2009, https://pubmed.ncbi.nlm.nih.gov/19678891/).
- It needs to be made clear when the terms CE and IBD are used in veterinary gastroenterology.
- The paper is missing a consequent differentiation between canine IBD, human IBD and experimental models of human IBD especially paragraphs 1 and 2.1. - 2.3.4. It looks like another author has taken care of paragraphs 2.3.5-3.3. since this part differentiates very well between species. In detail:
- Lines 55-56: Reference 3 concerns symptoms of human IBD but the sentence implies that the authors refer to clinical signs of canine IBD. It needs to be made very clear which information concerns dogs and which humans.
- Line 75: It needs to be stated that you refer here to healthy dogs.
- Line 86-87: Reference 9 refers to human Crohn's disease and no canine IBD
- Figure 1: Do the parts 1A-1E only refer to what is known from dog studies? It looks to me that there are also data from human IBD and experimental models involved. It should be made clear with references from which study the information was retrieved. You state yourself in lines 127-128 that "In dogs, a predominant immune profile for different chronic enteropathies has not been completely determined". This implies that some of the presented data is not from dogs. Please clarify.
- The legend of the figure needs an English revision, since parts of the sentences are difficult to understand
- Line 110: B) "Peptides ... regulate (plural) ... their expression" Question: The expression of what?
- Lines 112-113: "On the other hand" - is an expression favored by the authors but not always fitting. Please revise throughout the manuscript.
- Lines 115-116: "E) ... eosinophils in increased and F) ... mast cells ALSO decreased" - Question: Are mast cells also INcreased or are they DEcreased? Please clarify.
- Line 168: Intergrins - In IBD of what species?
- Line 179: Cytokines - In what IBD models?
- Line 188: In local wound sites - of what species?
4. Line 259: The sentence cannot be fully understood since it is missing what eosinophil derived factor activates mast cells.
5. Reference 75 is not about German shepherd dogs but about histiocytic ulcerative colitis (HUC) in Boxer dogs. HUC does not belong anymore to idiopathic IBD as stated above. (Mansfield et al. 2009) Please correct to the right reference or to the right content of the paper. Reference 75 is rightly cited in the table under "Innate immune cells and derived molecules" but wrongly under "T helper lymphocytes (CD4+)"
6. Lines 405-407: I did not understand this very long sentence. There is some word missing. Please revise.
7. I do strongly disagree that the dog is a promising model for studying human IBD. Canine IBD is substantially different from human IBD in clinical expression and in its major two types Crohn's disease and UC. Dogs, e.g., never require surgery to treat IBD. However, certain types of canine IBD show reference to human IBD as stated for canine eosinophilic GI disorders (ref 84). I suggest to be more cautious concerning the claim for spontaneous disease models and just write that some forms of canine IBD MAY HAVE POTENTIAL to serve as models.
Comments for minor revision:
- Keywords: Correct "Intestinal Bowel Disease" to "Inflammatory Bowel disease"
- Line 184 - Please add that the neutrophils were studies in duodenal mucosa
- Lines 192-194 - In what material have pANCA and NLR been measured?
- Line 196 - correct to "increase in the mucosa of the duodenum and colon"
- Line 199 - correct to "... calprotectin, another protein ..."
- Line 199-200 - correct to "... increases its expression in canine intestinal mucosa" when referring to reference [55]
Final Comment:
There have been some language issues that made it difficult to read/understand some parts of the paper. I do strongly recommend an English revision.
Author Response
Dear reviewer,
Thank you very much for your comments and corrections. We have worked to incorporate all of your recommendations. In addition, a native speaker has corrected the English throughout the manuscript.
Our responses to your review are detailed in the attached document.
Kind regards,
Daniela Siel.

Round 2
Reviewer 3 Report
The authors have appropriately addressed the comments of my first review. However, there are some issues that need to be corrected before recommending for publication.
Title
The title does not fully reflect the content of the article, since the focus of the article is not on potential biomarkers but rather on innate and adaptive immune response and the resulting suggestions for possible biomarkers. I suggest revision of the title to: "Elucidating the role of innate and adaptive immune response in the pathogenesis of canine chronic inflammatory enteropathy - A search for potential biomarkers"
Lines 22, 32, 54-56
With respect, I do disagree to use IBD/CE but suggest to use the term Chronic inflammatory enteropathy (CIE) throughout the entire article. From 2018-22, there has been minimum of 26 papers using the term CIE instead of IBD
(See: https://pubmed.ncbi.nlm.nih.gov/?term=canine%20%22chronic%20inflammatory%20enteropathy%22&sort=date).
When the term IBD has been wrongly used in the past it does not mean that the mistake should be continued. To be in line with the recent development in terminology, I strongly suggest to present the recent definitions for CE, CIE and IBD and use after this the term CIE throughout the article and not IBD/CE. I do also refer to your reference no. 100 (Heilmann, Steiner 2018).
A possible compromise could be to use IBD/CIE when the authors feel uncomfortable to replace the term IBD with CIE when citing authors who still have used the outdated term IBD. But when you decide to do so, please write also the reason why.
Line 58
When writing about human IBD, it is important to mention the predominant types of human IBD, Crohn's disease (CD) and Ulcerative Colitis (UC). The abbreviations CD and UC are used in several occasions throughout the manuscript but have not been defined at the start. This is important, since it determines the fundamental difference between human IBD and canine CIE. The change of canine IBD to canine CIE has also been implemented to avoid the misconception that canine CIE can serve as a one-to-one spontaneous disease model for human IBD.
Line 69
Please change "could" to "is considered"
Line 73
Please add that the treatment goals for human IBD are defined by STRIDE-II.
See: https://pubmed.ncbi.nlm.nih.gov/33359090/
Treatment in dogs aims currently only at clinical remission.
Line 79
Please rephrase to "intestinal microbiota disarrangement (dysbiosis),"
Line 114
Please delete "," and correct to "lead"
Line 133
Please correct to "Physiologically"
Line 177
Please correct to "as biomarkers"
Line 190
Please correct to "CIE", once you have defined CIE in the introduction.
Line 208
Please change to "boxer breed dogs with histiocytic ulcerative colitis (HUC)"
Line 211a
Please add: However, a later study revealed that HUC in boxer dogs is caused by enteroinvasive E. coli and can be successfully treated with enrofloxacin. Therefore, canine HUC is considered rather an infectious disease than belonging to the canine idiopathic CIE complex [4].
Line 211b
Please change to "A recent study in dogs of different breeds and with or without CIE"
Line 237
Please change "FRD" to "FRE"
Line 275
Please change to "human IBD patients"
Line 279
please change "in" to "of"
Line 290
Please change to "Classical pathway of complement activation"
Line 352
Here the abbreviation CD for Crohn's disease and UC for ulcerative colitis are used without previous mentioning of these forms of human IBD. Can be kept, if both disease types are mentioned before of if written in full here.
Line 387-8
Please write "modulating the microbiota diversity, maintaining the integrity of the intestinal barrier [127,128], and secreting a variety of antibodies.
Line 392
Please rephrase as "mucosa have been described in patients with CD" and delete "have been described" at the end of the sentence
Line 397-9
The sentence is too long and difficult to understand. Needs English revision.
Line 401-4
The sentence is too long and difficult to understand. Needs English revision.
Line 404-6
Please rephrase to: "In addition, a hypermethylation of TNFRSF13B and TNFRSF13C loci has been observed, possibly associated with a defect in IgA class switching [138,139]."
Author Response
Dear Reviewer:
Thank you very much for your second review, it has helped us a lot to improve our manuscript.
We have considered all your suggestions and requests. The changes performed in the revised version of the manuscript are listed point-by-point below.
With our best regards,
Daniela Siel
Galia Ramírez-Toloza

Round 3
Reviewer 3 Report
The authors have addressed all issues appropriately and I do not have any further comments except the notion that they should also use in the simple summary at line 24 the term "Chronic Inflammatory Enteropathy" instead of "Chronic enteropathy"
Author Response
Dear Reviewer,
Thank you very much again for your review, it has helped us a lot to improve our manuscript. We have considered your suggestion.
Best Regards
